# Combating Obesity: Harnessing the Synergy of Postbiotics and Prebiotics for Enhanced Lipid Excretion and Microbiota Regulation

**DOI:** 10.3390/nu15234971

**Published:** 2023-11-30

**Authors:** Yueming Zhao, Yaping Zheng, Kui Xie, Yanmei Hou, Qingjing Liu, Yujun Jiang, Yu Zhang, Chaoxin Man

**Affiliations:** 1Key Laboratory of Dairy Science, Ministry of Education, College of Food Science and Engineering, Northeast Agricultural University, Harbin 150030, China; yueming.zhao@ausnutria.com (Y.Z.); pingpingzyp0302a@163.com (Y.Z.); yujun_jiang@163.com (Y.J.); 2Ausnutria Dairy (China) Co., Ltd., Changsha 410000, China; kui.xie@ausnutria.com (K.X.); yanmei.hou@ausnutria.com (Y.H.); qingjing.liu@ausnutria.com (Q.L.)

**Keywords:** obesity, high-fat diet, fecal lipid, intestinal microbiota, Aiyisen compositions, metabolites

## Abstract

Obesity is a chronic metabolic disease that can be induced by a high-fat diet (HFD) and predisposes to a variety of complications. In recent years, various bioactive substances, such as probiotics, prebiotics, and postbiotics, have been widely discussed because of their good anti-lipid and anti-inflammatory activities. In this paper, soybean protein isolate was used as a substrate to prepare the postbiotic. Compound prebiotics (galactose oligosaccharides, fructose oligosaccharides, and lactitol) preparation Aunulife Postbiotics and Prebiotics Composition (AYS) is the research object. Weight loss and bowel movements in mice induced by a high-fat diet were studied. Moreover, qualitative and quantitative analyses of small-molecule metabolites in AYS were performed to identify the functional molecules in AYS. After 12 weeks of feeding, the weight gain of mice that were fed with high-dose AYS (group H) and low-dose AYS (group L) from 4 to 12 weeks was 6.72 g and 5.25 g (*p* < 0.05), both of which were significantly lower than that of the high-fat diet (group DM, control group) group (7.73 g) (*p* < 0.05). Serum biochemical analysis showed that TC, TG, and LDL-C levels were significantly lower in mice from the H and L groups (*p* < 0.05). In addition, the fecal lipid content of mice in the L group reached 5.89%, which was significantly higher than that of the DM group at 4.02% (*p* < 0.05). The study showed that AYS changed the structure of the intestinal microbiota in mice on a high-fat diet, resulting in a decrease in the relative abundance of *Firmicutes* and *Muribaculaceae* and an increase in the relative abundance of *Bacteroidetes*, *Verrucomicrobia*, and *Lactobacillus*. The metabolomics study results of AYS showed that carboxylic acids and derivatives, and organonitrogen compounds accounted for 51.51% of the AYS metabolites, among which pantothenate, stachyose, betaine, and citrate had the effect of preventing obesity in mice. In conclusion, the administration of prebiotics and postbiotic-rich AYS reduces weight gain and increases fecal lipid defecation in obese mice, potentially by regulating the intestinal microbiota of mice on a high-fat diet.

## 1. Introduction

Obesity, one of the major global health problems, causes an increased risk of diabetes, insulin resistance, cardiovascular disease, non-alcoholic fatty liver disease [1,2,3,4], and a range of chronic diseases. The prevalence of obesity is increasing, and obesity rates among children and adolescents have reached a historically high level. Although obesity can be controlled by increasing exercise and restricting diets, appropriate medication is needed for some patients. Studies have shown that taking phentermine, liraglutide, or orlistat for at least 12 months results in an overall weight loss of 2.9% to 6.8%. However, overdoses of these chemicals cause detrimental effects on the body [5]. Therefore, it is crucial to seek safe and effective methods of obesity prevention and treatment to maintain the metabolic balance of the body and to prevent and treat the occurrence of obesity symptoms.

In recent years, postbiotics have emerged as an effective tool for the treatment of obesity. Defined as “a preparation of inanimate microorganisms and/or their components that is beneficial to the host” by the International Scientific Association of Probiotics and Prebiotics (ISAPP) [6], postbiotics can be obtained by ultrasound, heating, solvent extraction, ultraviolet light, enzyme treatment, etc. It has probiotic functions such as immunomodulation, infection prevention, antioxidant, anti-inflammatory, blood pressure lowering, blood lipid lowering, and weight loss [7,8,9,10,11,12,13]. It was found that the gut microbiota’s release of the polyunsaturated fatty acid metabolite 10-hydroxy-cis-12-octadecenoic acid (HYA) could improve host resistance to high-fat diet-induced obesity [14]. In addition, Enterococcus faecalis and its metabolite, myristoleic acid, may reduce obesity by activating brown adipose tissue and forming beige fat. Therefore, postbiotics have the potential to regulate metabolism to improve obesity symptoms and are expected to be an effective alternative to other drug treatments to improve obesity.

Prebiotics also improve the energy and metabolic status of the body by increasing the abundance of beneficial microbiota in the intestine and having a hindering effect on the digestion and absorption of food [15]. The function and mechanism of well-known prebiotics such as galactose oligosaccharides, fructose oligosaccharides, human milk oligosaccharides (HMO), trans-oligo galactose (TOS), inulin, etc. are available elsewhere [16]. Among them, FOS is a class of naturally occurring oligosaccharides that are highly fermentable by the intestinal microbiota in the colon, thus promoting the growth of beneficial bacteria [17]. Studies have shown that oligofructose has a positive effect on blood lipids and that FOS can inhibit the accumulation of body fat induced by a high-fat diet and the absorption of dietary fat in the intestine [18]. In an animal experiment, a high-fat diet-induced mouse model was fed FOS for 6 weeks, which significantly reduced the Lees index and epididymal fat index in obese mice and restored the intestinal microbiota of the mice to a degree close to that of the normal group [19]. Another important prebiotic is GOS, a natural oligosaccharide that also has a positive effect on blood sugar and lipids in adults. GOS can improve host lipid homeostasis by promoting cholesterol catabolism and inhibiting obesity by accelerating the browning of white adipocytes and the thermogenesis of brown adipocytes [20]. Therefore, prebiotic supplementation can manage obesity-related comorbidities.

Even though the benefits of applying prebiotics or postbiotics are being investigated, there are few studies investigating the improvement of obesity in mice with postbiotic and prebiotic combinations and the underpinning mechanisms. In this study, the effect of a combination of postbiotic and prebiotic (AYS) on weight loss and defecation in high-fat diet-induced mice was investigated, and the effect of intestinal microbiota composition on obese mice was further analyzed.

## 2. Materials and Methods

### 2.1. Materials

Aunulife Postbiotics and Prebiotics Composition (AYS) was provided by Aunulife Biotechnology Co., Ltd. and stored at room temperature (Aunulife Biotechnology Co., Ltd., Changsha, China). Kits for TC (total cholesterol), TG (triglyceride), LDL-C (low-density lipid cholesterol), and HDL-C (high-density lipid cholesterol) were purchased from Shanghai Enzyme Link Biotechnology Co., Ltd. (Shanghai, China). Other reagents were obtained from Sino Pharma Co., Ltd. (Beijing, China).

### 2.2. AYS Non-Targeted Metabolic Assay

#### 2.2.1. Sample Pretreatment

A total of 80 g of AYS was dissolved in 200 μL of aqueous MP homogenate, vortexed for 60 s, and then added to 800 μL of methanol-acetonitrile solution. The vortex was continued for 60 s, and two repeated low-temperature sonications were performed for 30 min. The proteins were precipitated at −20 °C for 1 h, followed by centrifugation at 14,000 rpm at 4 °C for 20 min. The supernatant was freeze-dried and the sample was stored at −80 °C.

#### 2.2.2. Chromatography-Mass Spectrometry Analysis

The metabolic composition and metabolite structure of AYS were determined. The samples (AYS) were separated on an Agilent 1290 Infinity LC ultra-high performance liquid chromatography system (UHPLC) HILIC column. Column temperature: 25 °C; flow rate: 0.3 mL/min; mobile phase composition A: water + 25 mM ammonium acetate + 25 mM ammonia; B: acetonitrile. The samples were placed in a 4 °C autosampler during the entire analysis. To avoid the effects caused by fluctuations in the instrument detection signal, the samples were analyzed continuously in random order. QC samples are inserted in the sample queue to monitor and evaluate the stability of the system and the reliability of the experimental data. The samples were separated by UHPLC and analyzed by mass spectrometry using an Agilent 6550 mass spectrometer. The ESI source conditions are as follows: Gas Tem: 250 °C, Drying Gas: 16 L/min, Nebulizer: 20 psig, Sheath Gas Tem: 400 °C, Sheath Gas Flow: 12 L/min; Vcap: 3000 V; Nozzle Voltage: 0 V. Fragment: 175 V, Mass Range: 50–1200, Acquisition Rate: 4 Hz, Cycle Time: 250 ms. After the samples were assayed, the metabolites were identified using an AB Triple TOF 6600 mass spectrometer, and primary and secondary spectra of QC samples were acquired. The data obtained were used for the structural identification of the metabolites using the self-constructed MetDDA and LipDDA methods, respectively.

### 2.3. Animal Experimental Design

Forty male C57BL/6J mice (6 weeks, 17 ± 2 g) were provided by the Beijing Vital River Laboratory Animal Technology Co., Ltd. (Beijing, China). The animal experiment was approved by the Laboratory Animal Welfare and Ethics Committee of Northeast Agricultural University (#NEAUEC-2023-04-14) and was conducted according to the Guide for the Care and Use of Laboratory Animals (China). Mice were housed in 20 cages (2 animals per cage, *n* = 8 per group) and kept at 23 ± 2 °C and 55 ± 5% relative humidity, maintaining a 12-h light-dark cycle. In order to prevent the two mice in each cage from chewing on each other’s feces and competing for food, we used partitions to divide a cage into two areas so that each mouse could be fed separately. Mice were acclimatized for one week, during which all mice were fed a chow diet with free access to water, ensuring that mice were fed the same initial value of feed and drinking water (purified water). To evaluate the effect of this product on the ability of mice to lower lipid and fecal lipid content after one week of acclimatization feeding, the animal experiments were randomly divided into five experimental groups (*n* = 8 per group): the N group, the DM group, the H group (high-dose AYS), the L group (low-dose AYS), and the C group. Groups H, L, and C were gavaged daily for eight weeks, and the specific gavage methods and doses are shown in Table 1. High-fat diet: 38% normal diet, 20% lard, 10% soybean oil, 10% sucrose, 10% maltodextrin, 10% yolk powder, 1.8% cholesterol, 0.2% bile salt; Energy: 4.7 kcal/g, protein: 16%, fat: 42%, carbohydrate: 48%, and remaining nutrients: 4%. The chow was purchased from Shenyang Maohua Biotechnology Co., Ltd., (Shenyang, China) and the high-fat feed was purchased from Beijing Keao Cooperative Feed Co. (Beijing, China). Body weight measurements were performed weekly, and changes in mice were recorded. The mice had free access to chow and water throughout the feeding period.

At the end of the feed at week 12, mice were fasted overnight without water for 12 h. Eye blood was collected from mice by centrifugation at 3000 rpm/min for 20 min, and the supernatant was collected and stored in a −80 °C refrigerator. The necks were broken and executed, and the small intestine and other tissues were quickly frozen and stored in a −80 °C freezer. Fresh feces were collected weekly from the mice for intestinal microbiota analysis.

### 2.4. Changes in Body Weight and Fecal Quality of Mice

The body weight of the mice was measured weekly, the quality of the feces was measured every two weeks, and the mice were recorded to observe the number of fecal pellets and fecal status. Each mouse was photographed once a week to record its mental status and coat color status.

### 2.5. Determination of Biochemical Indicators in Serum

Whole blood was obtained by ophthalmoscopic blood sampling, rested at room temperature for 2 h, centrifuged at 4 °C for 10 min at 3000 r/min, and the supernatant was aspirated. Lipid levels in mice are measured according to kit instructions for total cholesterol (TC), total triglycerides (TG), low-density lipoprotein cholesterol (LDL-C), high-density lipoprotein cholesterol (HDL-C), and other lipid levels.

### 2.6. Determination of Fecal Lipid Content in Mice

Every two weeks, 9 g of mouse feces was selected from each group of mice, and three parallels were made. Five times the volume of extraction reagent (trichloromethane, methanol, and water = 2:2:1) was added, shook and mixed for 3 min, and centrifuged to obtain the lower chloroform layer. To extract adequately, after completing the first extraction, the upper aqueous phase was transferred to another glass test tube, adjusted to pH 1.5 or less by adding 0.2 mol/L HCl, acidified, and the lipid extraction steps were repeated. The first and second chloroform layers were then combined, blow-dried under nitrogen, and the mass determined. The formula is as follows:
$$\begin{array}{c} X=\frac{A}{B}\;✕\;100\%\\ \;X:\mathrm{faecal}\mathrm{lipid}\mathrm{content}\\ A:\mathrm{Lipid}\mathrm{mass}\mathrm{in}\mathrm{faeces}\\ B:\mathrm{Faecelquality} \end{array}$$


### 2.7. Intestinal Microbiota Analysis

The feces of each group of fed mice were collected and weighed at 200 mg in EP tubes, and the genomic DNA of the content samples was extracted using a DNA extraction kit. The V3-V4 high variant region of bacterial 16S rRNA was then amplified using universal primers 338F (5′-ACTCCTACGGGAGGCAGCA-3′) and 806R (5′-GGACTACHVGGGTWTCTAAT-3′). The amplified PCR products were tested by agarose gel electrophoresis and sequenced by an Illumina MiSeq sequencer to detect the composition and proportion of the major gut microbiota in each group of mouse samples.

### 2.8. Statistical Analysis

Each group of experiments was done in at least three parallels. GraphPad Prism 8.02 and Origin were used for plotting analysis, and SPSS 25.0 software was applied to perform a one-way ANOVA on all experimental data, with *p* < 0.05 representing significant differences.

## 3. Results

### 3.1. AYS Metabolite Analysis

Small molecule metabolites in AYS powder were analyzed by non-targeted metabolomic techniques. As shown in Table 2, though more than 15 categories of metabolites were identified in AYS, including carboxylic acids and derivatives, organonitrogen compounds were the most abundant metabolites identified in AYS, accounting for 51.51% of the total metabolites. Carboxylic acids and derivatives included Val-Val, Val-Asp, Val-Glu, Val-Tyr, Val-Pro, Val-Phe, Gly-Glu, Gly-Val, Gly-Arg, Thr-Glu, Thr-Leu, Ile-Glu, Ile-Met, Leu-Phe, Leu-Tyr, Pro-Thr, His-Phe, Arg-Ala, Ala-Lys, Tyr-Phe, Succinate, citrate, and betaine. Among them, betaine can reduce the methylation of the microsomal triglyceride transfer protein promoter by increasing genomic methylation in obese mice, which attenuates hepatic steatosis in mice [21]. Citrate prevents the deposition of lipids in the liver and adipose tissue in mice induced by a high-sugar, high-fat diet [22]. Organonitrogen compounds include Agmatine, Stachyose, Ile-Lys, Pantothenate, Maltotriose, Sucrose, Ribitol, Diacetyl, Maltotriose, and D-Mannose. Among them, hydrolase binding to berberine in vivo ameliorates disorders of glucolipid metabolism in mice with spontaneous type 2 diabetes. Pantothenate reduces obesity by increasing energy expenditure in C57BL/6J mice fed a high-fat diet-induced diet. Therefore, the non-targeted metabolomic analysis demonstrated that AYS metabolites contain components that reduce weight loss in mice induced by a high-fat diet.

### 3.2. The Combination of Prebiotics and Postbiotics Reduces the Body Weight Gain in Mice

From the modeling period to the end of feeding, the body weight of each group of mice was recorded and analyzed (Figure 1). The initial body weight values of all five groups of mice were 17–19 g. There was no significant difference between the groups. After 12 weeks of continuous feeding, all five groups of mice showed an overall increase in body weight. From week 2 to week 4 of the experiment, the body weight of mice in the high-fat diet groups was higher than that of mice on a normal diet, reaching 22.40 ± 0.23 g in the DM group at week 4, 13.50% higher than the body weight of group N at week 4. From week 5 to week 12 of the test, the mice in the DM group continued to gain weight; however, the mice in groups C, H, and L had a slower rate of weight gain, and the mice gained less weight than those in the DM group. At the end of the twelfth week, the weight gain compared to the weight at the fourth week was 5.64 g, 6.72 g, and 5.25 g in the C, H, and L groups, respectively, which was lower than the 7.73 g in the HFD group (*p* < 0.05). Therefore, the administration of AYS prebiotics and postbiotics alleviates weight gain in mice, and the best results are achieved at low doses.

### 3.3. Effect on Fecal Lipid Content and Defecation of Mice

We further investigated the effect of AYS on the defecation of obese mice. As shown in Figure 2a, each group had normal bowel movements from the modeling period to the end of feeding. Mice fed with different doses of AYS promoted defecation compared to the DM group at 4–12 weeks. Low doses of AYS are more effective in promoting bowel movements. The number of pellets defecated by mice at the end of week 12 is shown in Figure 2b, which is consistent with the results of fecal quality in mice. Therefore, it indicates that gavage of AYS can promote defecation in mice.

As shown in Figure 2c, the lipid content in the feces of the drug-administered mice in group C (positive control) tended to be stable at 3–5%. The fecal lipid content in the DM group was higher than in the N group, and the fecal lipid content ratio in the AYS groups was significantly higher than in the DM group (*p* < 0.05). A significantly higher ratio (*p* < 0.05) of fecal lipid was observed in mice in the L group than in the H group after 6–12 weeks of feeding. This indicates that mice fed low-dose AYS can promote oil excretion.

### 3.4. Effects on Serum Levels of TC, TG, LDL-C, and HDL-C in Mice

Previous studies have shown that the increase of TC, TG, and LDL-C content and the decrease of HDL-C content in serum are related to the occurrence of nonalcoholic fatty liver disease, atherosclerosis [23], and cardiovascular disease symptoms [24]. To investigate whether AYS (prebiotics and postbiotics) regulate the metabolism of mice, TC, TG, LDL-C, and HDL-C in the serum of mice were measured and compared between different groups.

After ten weeks of high-fat diet feeding, the serum levels of TC, TG, LDL-C, and HDL-C in the DM group mice were significantly higher than those in the N group (*p* < 0.05, Figure 3). In contrast, TC, TG, and LDL-C levels significantly decreased and HDL-C levels significantly increased in the C, H, and L groups when compared with the DM group (*p* < 0.05). Mice in the L group showed the best-preventing effects among the experimental groups; serum TC (3.22 ± 0.13 mmoL/L), TG (1.26 ± 0.17 mmoL/L), and LDL-C (1.05 ± 0.05 mmoL/L) levels were significantly lower (*p* < 0.05) and HDL-C (2.73 ± 0.17 mmoL/L) levels were significantly higher (*p* < 0.05) when compared with the DM group. Similar to the results of Fu et al., which show that banana-resistant starch modulates the normalization of TC, TG, LDL-C, and HDL-C levels in the serum of obese rats induced by a high-fat diet [25], our results suggest that low doses of AYS can better maintain normal lipid levels in the body by improving serum levels of TG, TC, LDL-C, and HDL-C.

### 3.5. Effect on Intestinal Microbiota Composition

#### 3.5.1. Alpha Diversity Analysis

A high-sugar and high-fat diet can alter the composition and diversity of the intestinal microbiota, influence the physiological functions of the host, and have a crucial impact on the human body. Long-term dietary intervention may be a safe and effective way to prevent and treat obesity. Alpha diversity reflects the number and diversity of microbial species in mouse feces and is commonly measured by the Chao1 index, the Shannon index, and the Simpson index [26]. The level of this index represents the richness and homogeneity of the microbial community in the sample. As shown in Figure 4, the Chao1 index, Shannon index, and Simpson index were significantly lower in the DM group compared to the N group (*p* < 0.05), indicating that a high-fat diet reduces the abundance and diversity of intestinal microbiota in mice. On the other side, the Chao1 index, Shannon index, and Simpson index were significantly higher in the C, H, and L groups compared with the DM group (*p* < 0.05), which indicates that the dietary intervention had a positive effect on maintaining intestinal microbiota diversity in mice. In addition, low-dose administration of AYS is the most efficient dose to restore the richness and diversity of the intestinal microbiota in high-fat mice.

#### 3.5.2. Beta Diversity Analysis

Beta diversity analysis was used to investigate the effects of different interventions on the intestinal microbiota of mice in five groups: N, DM, C, H, and L. The results are shown in Figure 5. The beta diversity analysis mainly includes PCA, PCoA analysis, NMDS analysis, etc. Differences between individuals or groups can be observed using PCA. The closer the distance between two points on the graph, the more similar the species composition. Compared with the DM group, the three dietary intervention groups showed a higher aggregation of sample sites and were closer to the N group overall. The results suggest that feeding mice with low-dose AYS in the L group can restore the intestinal flora and improve the stability of the community structure of the intestinal microbiota in high-fat diet-fed mice.

#### 3.5.3. Analysis of Intestinal Microbiota Composition

##### Composition of Intestinal Microbiota at Phylum Level

A long-term high-fat diet will cause a disorder of the internal environment of the body’s intestines and then affect the composition of the intestinal microbiota. An analysis of microbial composition at the phylum level was performed to analyze the differences among groups N, DM, C, H, and L at the phylum level. As shown in Figure 6a, the most abundant bacteria in all five groups are *Firmicutes, Bacteroidota*, *Verrucomicrobia*, *Acidobacteriota*, *Proteobacteria*, *Gemmatimonadota*, *Patescibacteria*, *Cyanobacteria*, and *Deferribacteia*. The relative abundance of *Firmicutes* increased and the relative abundance of *Bacteroidetes* decreased in the DM mice compared with the N group. The dietary intervention in the C, H, and L groups reversed this trend and converged in the N group, which is consistent with the findings of Dong et al. [27]. In addition, some reports demonstrate that obese mice have a higher *Firmicutes/Bacteroidetes* ratio than lean mice [28]. *Verrucomicrobia* is closely associated with the development of diabetes and obesity, and induction by a high-fat diet resulted in a decrease in the relative abundance of *Verrucomicrobia* in mice in the DM group, whereas dietary interventions in the C, H, and L groups increased the relative abundance of *Verrucomicrobia*. Therefore, the dietary interventions in the C, H, and L groups could restore the intestinal microbiota of mice, which is disrupted by a high-sugar and high-fat diet.

##### Composition of Intestinal Microbiota Based on Genus Level

Genus level-based microbial community composition analysis was performed to analyze the differences in the composition of the intestinal microbiota of mice in groups N, DM, C, H, and L at the genus level (Figure 6b). At the genus level, *Akkermansia*, *Muribaculaceae_unclassified*, *Lactobacillus*, *Lachnospiraceae_NK4A136_group*, *Ruminococcaceae_UCG-014*, *Prevotellaceae_UCG-001*, *Coriobacteriaceae_UCG-002*, *Bifidobacterium*, and *Enterorhabdus* are the most abundant genus in all groups. The relative abundance of *Muribaculaceae*, which has been reported to be associated with obesity [29], increased, and the relative abundance of *Lactobacillus* decreased in the DM group compared to the N group, whereas dietary interventions in the C, H, and L groups resulted in a convergence of the relative abundance of *Muribaculaceae* and *Lactobacillus* to the N group level. *Akkermansia* found in healthy human feces was associated with obesity [30]. Reducing the relative abundance of *Akkermansia* can accelerate the development of obesity [31]. Compared with the DM group, the dietary intervention in the C, H, and L groups increased the relative abundance of *Akkermansia*.

##### LEfSe Analysis of Mice Gut Microbiota

LEfSe analysis is able to look for biomarkers that are statistically different between groups, with different colors indicating species in different subgroups. LEfSe analysis was used to analyze the main changes in the intestinal microbiota of mice after the induction of a high-fat diet, as shown in Figure 7. As seen in the figure, the relative abundance of *Akkermansia*, *Verrucomicrobiae*, *Clostridioides*, and *Peptostreptococcaceae* was significantly increased in group L and significantly different from the control group. Studies have shown that an increase in *Akkermansia* is strongly associated with metabolic diseases such as obesity and type 2 diabetes. Taken together, the results suggest that the dietary intervention in the C, H, and L groups can regulate the intestinal microbiota and alleviate the dysbiosis of the intestinal microbiota caused by long-term intake of a high-fat diet in mice by changing the relative abundance of *Akkermansia*, *Muribaculaceae*, and *Lactobacillus*.

## 4. Discussion

Obesity is a phenomenon in which the body deviates from normal energy homeostasis due to the abnormal accumulation of fat [32]. At present, obesity is widely prevalent throughout the world, and a high-fat diet and a sedentary lifestyle are the main causes of metabolic diseases such as obesity, associated with a decrease in microbial diversity. Recent studies have shown that regulating intestinal microbiota disorders through dietary interventions to protect the intestinal barrier can improve obesity and other metabolic diseases [33]. Therefore, it is necessary to prevent or improve the occurrence of obesity through dietary interventions. In this study, a high-fat diet was used to construct an obese mouse model, which provides a theoretical basis for the treatment of obesity based on postbiotics and probiotics and provides ideas for the application of AYS to products with therapeutic effects on obesity.

The introduction of metabolomics technology provides new methods with which to trace the origin and authenticity of agricultural products, reveal changes in nutrient composition during growth and storage, study the mechanism of nutrient function, and provide new strategies for the optimal adjustment of the dietary structure. Metabolomics is a technique for the comprehensive qualitative and quantitative analysis of all endogenous small-molecule metabolites in an organism under specific conditions [34]. The raw data collected by mass spectrometry is used to identify metabolites by database search and molecular network technology to produce more accurate results. In this study, the metabolites of AYS were identified, and AYS was found to contain succinate, citrate, betaine, agmatine, stachyose, ile-Lys, pantothenate, maltotriose, ribitol, dicetyl, D-mannose, and dipeptide metabolites. Among other things, succinate is produced by intestinal microorganisms and depletion of succinate levels in the body is associated with the regulation of obesity, diabetes, and other symptoms. Studies have shown that oral administration of Odoribacter lanes depletes succinate levels in mice and reduces levels of triglycerides and uric acid, thereby improving the symptoms of obesity [35]. Stachyose is an oligosaccharide that regulates the composition of the intestinal microbiota of animals. Studies have shown that supplementation with non-digestive stachyose enhances the absorption of soy isoflavones and thus prevents weight gain and fat accumulation in HFD-induced C57/BL6 mice [36]. In addition, AYS metabolites contain betaine, a natural compound found in foods that has a potential role in regulating lipid metabolism. Research showed that betaine supplementation could promote hepatic lipid metabolism in mice by increasing metabolic exercise in obese mice [37]. In addition, increasing the maternal intake of betaine during lactation resulted in reduced obesity and an increased relative abundance of *Akkermansia* in the intestine of mouse offspring throughout adulthood [38]. This is consistent with the findings that feeding AYS increased the relative abundance of *Akkermansia* in the intestinal microbiota of HFD-induced mice. However, we need to continue to determine what substances in AYS play a major role in obesity in mice. Thus, AYS metabolites contain components that can reduce weight gain due to HFD in mice.

The main characteristics of high-fat diet-induced obesity in mice are the increase in body weight, the increase in blood lipid and cholesterol levels, specifically in serum TG, TC, and LDL levels, and the decrease in HDL levels [39]. In this study, mice were fed a high-fat diet, and the results showed that after the AYS diet intervention, the body weight of mice in the high-dose and low-dose AYS groups was lower than that of the DM group, in which the blood lipid levels (TC, TG, and LDL-C) were increased and HDL-C levels were decreased, indicating that the mice were successfully modeled. After different doses of AYS gavage, the serum levels of TG, TC, and LDL in mice showed different trends of decrease, and HDL levels showed a trend of increase; the effect was more significant in the low-dose group. It was found that GOS fed to hypercholesterolemic female rats significantly reduced serum TC, TG, and LDL-C levels in rats, which is consistent with the results of this experimental study [40]. Therefore, AYS can improve the symptoms of obesity by regulating blood lipid and cholesterol levels.

With an increasing standard of living, diet and health issues are widely discussed. In recent years, it has been found that a high-fat diet induces abnormal bowel function in obese individuals [41]. Reasonable dietary interventions can effectively regulate the structure of the intestinal microbiota and achieve an effective improvement in abnormal bowel function. In the present study, the lipid content in the feces of the low-dose AYS group was significantly higher than that of the DM group, and in addition, the number and weight of defecations were significantly increased in the low-dose AYS group. Wu et al. showed that feeding HFD-induced mice with monoclonal antibody blockade of long fragment neurotensin increased their fecal lipid content, similar to the results of the present study [42]. In addition, it was shown that feeding the outer bran fraction of rice increased the fecal count, weight, and lipid content of rats to prevent lipid accumulation [43]. Thus, AYS promotes the excretion of feces and its lipid content in HFD mice.

The intestinal microbiota plays a key role in the regulation of host metabolism. The long-term intake of a high-fat diet can cause disturbances in the organism’s intestinal environment and reduce the diversity of the intestinal microbiota. In this experiment, the alpha diversity index of mice in the DM group decreased, whereas the high-dose and low-dose AYS groups increased the alpha diversity of mouse intestinal microbiota, indicating that AYS can alter the relative abundance and diversity of intestinal microbiota in mice on a high-fat diet. At the gate level, the high-dose and low-dose AYS groups fed AYS improved the dysbiosis of intestinal microbiota in mice induced by a high-fat diet by decreasing the relative abundance of *Firmicutes* and *Verrucomicrobia* and increasing the relative abundance of *Bacteroidetes*. It was found that the intervention of prebiotic GSF in combination with cellular components of *Lactobacillus* kefir reduced energy intake in mice by modulating the composition of their intestinal microbiota, which is consistent with the results of the present study [44]. At the genus level, the relative abundance of *Muribaculaceae* increased and that of *Lactobacillus* and *Akkermansia* decreased in the DM group, whereas dietary intervention in the high-dose and low-dose AYS groups reversed this trend. *Muribaculaceae* is one of the dominant bacteria in the intestinal microbiota that breaks down carbohydrates. Research showed that the proliferation of *Muribaculaceae* was promoted by feeding cooked red beans [45], and in turn, the relative abundance of *Muribaculaceae* in the DM group was higher than in the N group, similar to the results obtained in the present experiment. *Akkermansia* is a Gram-negative anaerobic bacterium that shows a negative association with diseases such as obesity [46]. *Akkermansia* was less abundant in the obese population compared to the healthy population, and the obese population showed a significant reduction in blood glucose parameters and, thus, obesity after *Akkermansia* supplementation [47]. Therefore, AYS can improve the symptoms of obesity by regulating the imbalance of the gut microbiota caused by HFD. The study showed that the effects of AYS on body weight, blood lipid level, defecation ability, and intestinal microbiota were more pronounced in the low-dose group. It is possible that the low dose of AYS can be more fully utilized by the mice for its efficacy because it is more helpful for their digestion and absorption. The high dose of AYS may be excreted because the dose is too much for the mice to be fully absorbed and utilized, so the low dose of AYS is more effective in mice. However, this study is still in its early stages, and the molecular mechanism of the combined intervention of prebiotics and postbiotics in the clinical prevention or treatment of lipid lowering and related metabolic disorders needs to be further investigated.

## 5. Conclusions

In this study, a non-targeted metabolomics approach was used to analyze the derived AYS metabolites. By constructing a high-fat-induced mouse model, the body weight of mice in the L group was significantly reduced after 10 weeks of feeding, indicating that the administration of low-dose AYS could effectively control obesity in mice and significantly increase the fecal lipid content. Second, the administration of low doses of AYS resulted in a significant decrease in TC, TG, and LDL-C levels and a significant increase in HDL-C levels in mice. In addition, the administration of low-dose AYS interventions could improve the intestinal microbiota of high-sugar, high-fat diet-induced mice by altering the relative abundance of *Firmicutes*, *Bacteroidota*, and *Verrucomicrobia* to a certain extent. This research has important implications for AYS’s ability to improve obesity symptoms and for future research on related products.

## Figures and Tables

**Figure 1 nutrients-15-04971-f001:**
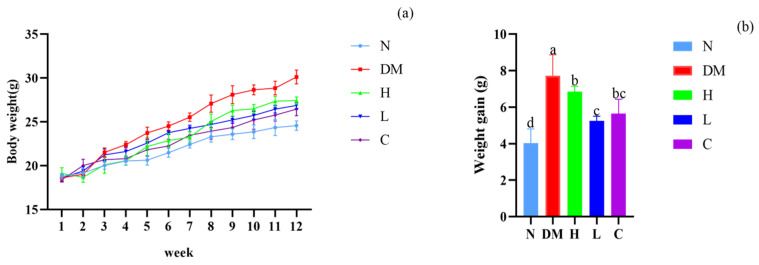
(**a**) Change in body weight of mice and (**b**) body weight gain from a successful mouse model (week 4) to the end of feeding (week 12). Different letters denote significant differences between different groups of mice (*p* < 0.05). N: normal diet; DM: diet model; H: high-dose AYS; L: low-dose AYS; and C: control.

**Figure 2 nutrients-15-04971-f002:**
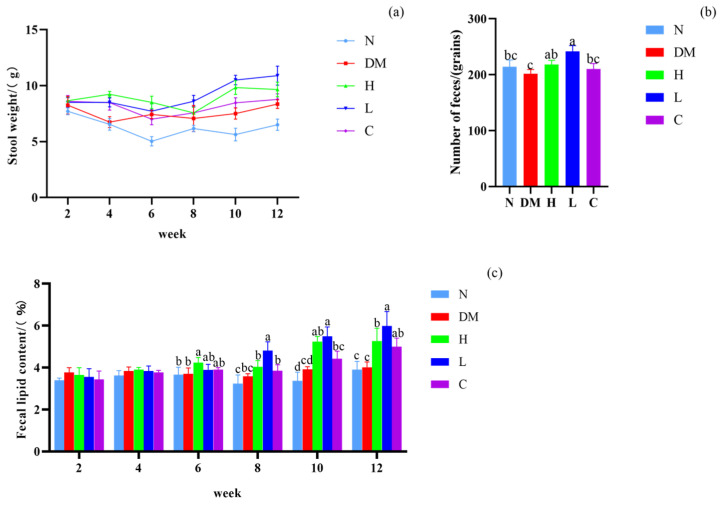
(**a**) Changes in fecal quality of mice; (**b**) number of fecal pellets in mice from the start of feeding until the end of week 12; and (**c**) changes in fecal lipid content of mice from week 2 to the end of week 12 of feeding. Abbreviations for each group are shown in Figure 1. Different letters denote significant differences between different groups of mice (*p* < 0.05). N: normal diet; DM: diet model; H: high-dose AYS; L: low-dose AYS; and C: control.

**Figure 3 nutrients-15-04971-f003:**
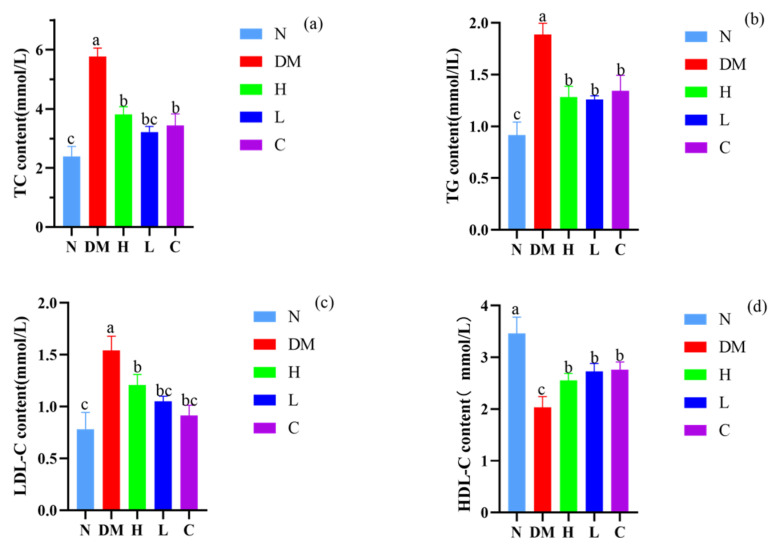
(**a**) Changes in serum total cholesterol (TC) content in mice; (**b**) changes in serum total triglyceride (TG) content in mice; (**c**) changes in serum low-density lipoprotein cholesterol (LDL-C) levels in mice; (**d**) and changes in serum high-density lipoprotein cholesterol (HDL-C) content in mice. Different letters denote significant differences between different groups of mice (*p* < 0.05). N: normal diet; DM: diet model; H: high-dose AYS; L: low-dose AYS; and C: control.

**Figure 4 nutrients-15-04971-f004:**
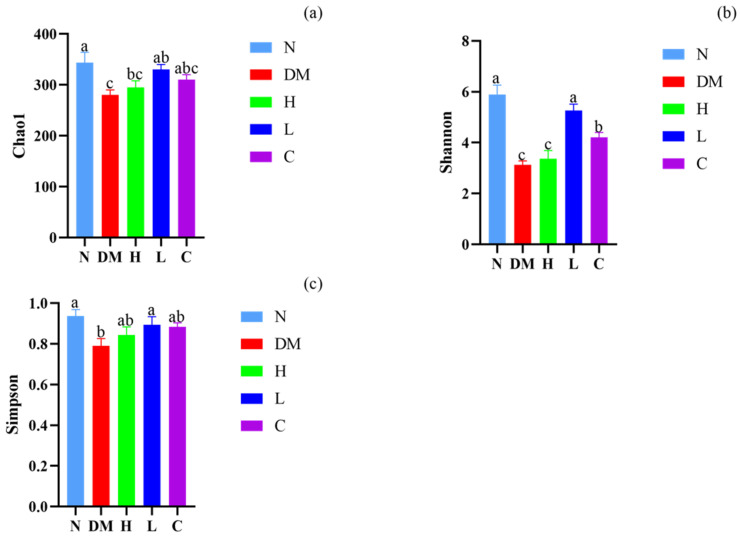
(**a**) Changes in the Chao1 index of alpha diversity in mouse intestinal microbiota; (**b**) changes in the Shannon index of alpha diversity in mouse intestinal microbiota; and (**c**) changes in the Simpson index of alpha diversity in mouse intestinal microbiota. Different letters denote significant differences between different groups of mice (*p* < 0.05). N: normal diet; DM: diet model; H: high-dose AYS; L: low-dose AYS; and C: control.

**Figure 5 nutrients-15-04971-f005:**
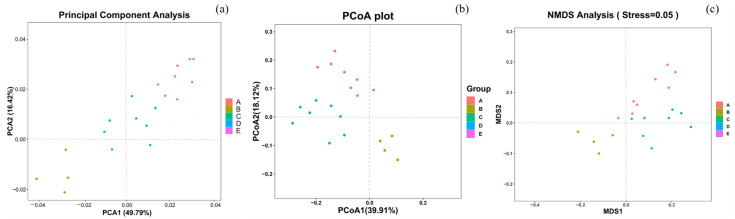
(**a**) Principal component analysis; (**b**) principal coordinate analysis; and (**c**) MDS analysis. Note: A represents group N, B represents group DM, C represents group C, D represents group H, and E represents group L.

**Figure 6 nutrients-15-04971-f006:**
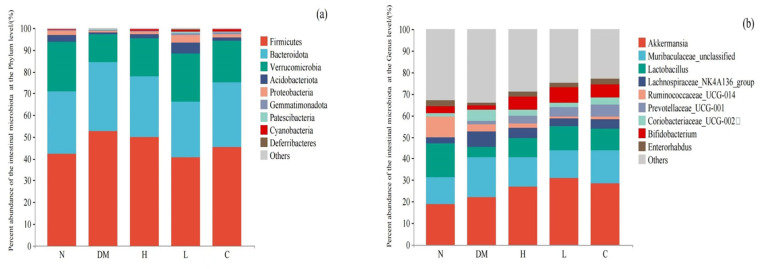
(**a**) Differences in the composition of mouse intestinal microbiota at the phylum level, and (**b**) differences in the composition of mouse intestinal microbiota at the genus level. N: normal diet; DM: diet model; H: high-dose AYS; L: low-dose AYS; and C: control.

**Figure 7 nutrients-15-04971-f007:**
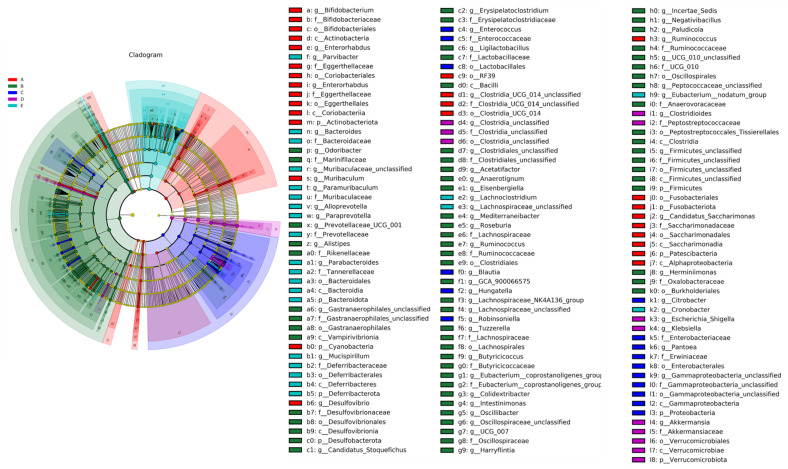
LEfSe analysis of mice gut microbiota. A represents group N, B represents group DM, C represents group C, D represents group H, and E represents group L.

**Table 1 nutrients-15-04971-t001:** Treatment for experimental mice.

Group	Diet	Treatment
N	Normal diet	0.2 mL PBS per day
DM	High-fat diet	0.2 mL PBS per day
H	High-fat diet	200 μL of AYS per day
L	High-fat diet	100 μL of AYS per day
C	High-fat diet	200 μL postbiotics of Yicuizhi dilute solution per day

**Table 2 nutrients-15-04971-t002:** Composition of AYS metabolites.

No.	Compounds	M/z	Rt (s)	B2	Class
1	Val-Val	217.1537	224.884	134,779.7	Carboxylic acids and derivatives
2	Val-Asp	233.1115	382.094	89,946.17	Carboxylic acids and derivatives
3	Val-Glu	247.1313	402.493	2501.2	Carboxylic acids and derivatives
4	Val-Tyr	281.1486	231.593	173,349.6	Carboxylic acids and derivatives
5	Val-Pro	215.1373	261.452	176,255.3	Carboxylic acids and derivatives
6	Val-Phe	265.1537	190.596	424,802.5	Carboxylic acids and derivatives
7	Gly-Glu	205.0854	440.611	28,407.88	Carboxylic acids and derivatives
8	Gly-Val	175.1071	390.344	11,429.14	Carboxylic acids and derivatives
9	Gly-Arg	232.1391	448.69	25,774.23	Carboxylic acids and derivatives
10	Thr-Glu	231.0964	306.569	20,522.67	Carboxylic acids and derivatives
11	Thr-Leu	233.1492	250.772	801,098.3	Carboxylic acids and derivatives
12	Ile-Glu	261.1436	359.036	30,986.21	Carboxylic acids and derivatives
13	Ile-Met	263.1417	193.436	280,259.7	Carboxylic acids and derivatives
14	Leu-Phe	279.1695	177.737	412,315.3	Carboxylic acids and derivatives
15	Leu-Tyr	295.1643	212.834	629,953.3	Carboxylic acids and derivatives
16	Pro-Thr	199.1063	119.86	64,398.15	Carboxylic acids and derivatives
17	His-Phe	285.1339	148.378	478,955.4	Carboxylic acids and derivatives
18	Arg-Ala	246.154	449.38	33,634.96	Carboxylic acids and derivatives
19	Ala-Lys	218.1479	437.541	166,954.3	Carboxylic acids and derivatives
20	Tyr-Phe	311.1371	39.745	26,125.42	Carboxylic acids and derivatives
21	Succinate	117.0193	432.639	11,464.94	Carboxylic acids and derivatives
22	Citrate	191.02	553.862	334,888.1	Carboxylic acids and derivatives
23	Betaine	118.086	274.181	7,530,456	Carboxylic acids and derivatives
24	Agmatine	131.1272	450.11	96,303.9	Organonitrogen compounds
25	Stachyose	667.227	488.158	210,245.2	Organooxygen compounds
26	Ile-Lys	260.1943	354.886	37,147.84	Organooxygen compounds
27	Pantothenate	220.1162	276.731	24,141.14	Organooxygen compounds
28	Maltotriose	503.1593	504.935	7323.386	Organooxygen compounds
29	Pantothenate	218.1041	277.019	150,933	Organooxygen compounds
30	Sucrose	341.1078	488.116	41,952.1	Organooxygen compounds
31	Ribitol	133.051	199.833	1,769,454	Organooxygen compounds
32	Diacetyl	87.04301	177.737	136,216.7	Organooxygen compounds
33	Maltotriose	522.198	494.367	65,036.57	Organooxygen compounds
34	D-Mannose	239.0775	296.488	78,534.97	Organooxygen compounds
35	Hypoxanthine	137.0436	258.532	721.8929	Imidazopyrimidines
36	Adenine	134.0478	158.426	2,083,855	Imidazopyrimidines
37	Hypoxanthine	135.0314	167.125	862,657.7	Imidazopyrimidines
38	Uric acid	167.0213	329.316	11,690.18	Imidazopyrimidines
39	Xanthine	151.0262	214.702	242,308.8	Imidazopyrimidines
40	Cytosine	112.0496	196.825	315,753.4	Diazines
41	Uracil	113.0333	159.348	35,649.18	Diazines
42	Thymine	125.0354	72.761	549,390.5	Diazines
43	Uracil	111.0196	284.668	19,261.86	Diazines
44	Orotate	155.0098	248.47	270,828.3	Diazines
45	Nicotinate	122.0245	227.3765	33,340.51	Pyridines and derivatives
46	Pyridoxine	170.0804	132.01	165,434.2	Pyridines and derivatives
47	Nicotinate	124.0377	232.263	90,057.04	Pyridines and derivatives
48	Benzoic acid	121.0293	15.862	22,075.6	Benzene and substituted derivatives
49	Tyramine	138.0904	214.194	25,018.14	Benzene and substituted derivatives
50	Hippuric acid	178.0505	193.264	609,644	Benzene and substituted derivatives
51	Palmitic acid	255.2328	49.642	6,242,620	Fatty Acyls
52	Linoleic acid	279.2339	36.733	1,609,868	Fatty Acyls
53	Heptadecanoic acid	269.2489	48.362	181,756.1	Fatty Acyls
54	Riboflavin	377.1541	250.772	3816.888	Pteridines and derivatives
55	Riboflavin	435.1591	262.95	3317.16	Pteridines and derivatives
56	1,2-dioleoyl-sn-glycerol-3-phosphatidylcholine	786.5968	95.831	85,056.19	Glycerophospholipids
57	sn-Glycerol 3-phosphoethanolamine	216.0617	393.803	59,884.73	Glycerophospholipids
58	Norharmane	169.0754	72.333	164,364.3	Indoles and derivatives
59	Indolelactic acid	204.0666	159.156	112,7278	Indoles and derivatives
60	Ile-Pro	229.1539	272.911	246,708.1	Pyrimidine nucleosides
61	Deoxythymidine 5′-phosphate (dTMP)	321.0482	411.011	10,607.32	Pyrimidine nucleotides
62	Cholesteryl sulfate	465.3023	26.694	165,882.3	Steroids and steroid derivatives
63	Cholic acid	426.3198	226.884	5626.847	Steroids and steroid derivatives
64	(S)-2-Hydroxyglutarate	147.0304	392.132	111,556.1	Hydroxy acids and derivatives
65	Dopamine	136.0745	255.082	9297.395	Phenols
66	Bestatin	350.204	200.345	35,719.13	Peptidomimetics

## Data Availability

Data in the project are still being collected, but all data used in the study is available by contacting the authors.

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
