# Peer review of "Combating Obesity: Harnessing the Synergy of Postbiotics and Prebiotics for Enhanced Lipid Excretion and Microbiota Regulation"

_nutrients, 2023, doi:10.3390/nu15234971_

Round 1

Reviewer 1 Report

Comments and Suggestions for Authors

The authors evaluate the impact of postbiotic and prebiotic combinations on improving obesity in mice and the mechanisms.

The proposal is to analyze the effect of a combination of postbiotic and prebiotic in high-fat diet-induced mice. The parameters the authors analyzed are weight loss and defecation and the impact of intestinal microbiota composition on obese mice.

Whole text:

- review for typos and improve English style

Methods:

- clarify how the authors controlled the mice ingesting debris or shredded tissue paper when they were kept overnight with no food.

- are the mice kept alone in the cages? Could the authors clarify it and discuss the implications?

Results:

- the authors present the % of the abundance of phylum and the genus level. Is that relative or total abundance?

Discussion:

- limitations need to be included. Please consider including the limitations of the protocols/procedures and the analysis.

Comments on the Quality of English Language

Whole text:

- review for typos and improve English style (grammar and shorter sentences)

Reviewer 2 Report

Comments and Suggestions for Authors

An interesting manuscript “Combating Obesity: Harnessing the synergy of postbiotics and prebiotics for enhanced lipid excretion and microbiota regulation”.

Abstract

Line 13 please change caused to which can be induced. A high fat is not the only cause of obesity.

Methodology Section;

It is unclear in the methods section of the manuscript how the postbiotics and prebiotics were administered to the mice. In other areas of the manuscript there is mention of gavage. Were the postbiotics and prebiotics delivered via gavage to the mice? If so, please make this clear in the methodology section.

Line 127, please change normal diet to either chow or standard chow diet. This should be consistent throughout the manuscript.

Did the mice have ad libitum access to food and water, or was it only water?

Line 126, was there a temperature and humidity range?

What was the animal ethics approval number?

Were all procedures involved performed in accordance with National guidelines?

Line 136; change to chow rather than basic feed, please be consistent throughout manuscript with terminology.

Line 139, refers to test, what is it meant by test?

Line 143, please change refrigerator to freezer

Was food intake monitored?

Line 152, each mouse was photographed, was this recorded via video or a photo of the mouse, was this done every 2 weeks also?

Were fat pads obtained from the mice? Was the definition of obese mice a higher body weight than the chow fed mice? Was body composition measured to look at the body fat composition?

Results Section

Significance should have symbols and be highlighted on Figure 1 between groups.

Line 272 change (C) to (c) to be consistent with the other lower case letters.

Figure 5 is difficult to read, can this be made clearer to the reader?

Discussion Section;

The L group is referred to quite often, this is fine in the methodology section but in the discussion section please change to the actual treatment type, so that it is easier for the reader to follow without having to refer back to the methods section.

Line 365, please change irregular lifestyle to sedentary lifestyle.

Comments on the Quality of English Language

Comments included above.

Reviewer 3 Report

Comments and Suggestions for Authors

Dear editor, first of all I want to thank you for the possibility of reviewing this article. It is great work that can open avenues of research in the treatment of obesity through the use of prebiotics and probiotics. At first I see the article as too dense and I think it should be divided into two. On the one hand, the anthropometric and analytical repercussions and on the other hand, placing more emphasis on certain aspects in the study of the fecal microbiota. But it is a question that I leave to the editor and the authors. In order to improve the presentation, I think that the graphs should be represented in colors instead of filled, always maintaining the same colors for each group, since the comparisons would be more visual. Finally, Figure 2c would be much more illustrative if It will be carried out in a similar way to Figure 2a.

Round 2

Reviewer 1 Report

Comments and Suggestions for Authors

- Based on the answers regarding how the mice were housed in cages, how did the authors control mice eating feces from each other? Please explain and discuss the implications of this to the results and include your consideration in the paper.

Reviewer 2 Report

Comments and Suggestions for Authors

Thank you to the reviewers for the changes that they made. They have done a lot of work in amending the manuscript. I have one further comment in response to Response 2. Please see below in bold.

Response 2: Thank you for your kind suggestion and question. During the feeding of the mice, we gave the prebiotics and the postbiotics to the mice by gavage. We have made an addition on page 3, paragraph 3, line 136 of the manuscript. “Groups H, L, and C were gavaged continuously for eight weeks, and the specific gavage methods and doses are shown in Table 1.”

Please change continuously to daily.

Comments on the Quality of English Language

Please see above.
